# Case Study on Spatial Mismatch between Multivariate and Student-Teacher Rate in U.S. Public School Districts

Xiu Wu [1,2] and Jinting Zhang [1,3,*]

1 Key Laboratory of Geographic Information System of MOE at Wuhan University, Wuhan 430070, China; x_w10@txstate.edu
2 Department of Geography and Environmental Studies, Texas State University, San Marcos, TX 78666, USA
3 School of Resource and Environmental Science, Wuhan University, Wuhan 430070, China
* Correspondence: whuzjt@whu.edu.cn; Tel.: +86-13387509245

**Abstract:** An important aspect of educational equity is the balance between students and teachers in the general education system. To address the need for a sustainable, balanced, and reliable supply of high-quality STEM teachers for public school districts, this research aims to illustrate the spatial dynamics of student-teacher balance in the impact of teacher salary, school poverty, transportation, and environmental factors from 2015 to 2020, Data were collected to create a multivariate evaluation via Analytic Hierarchical Process (AHP), Compromise Programming (CP), weight linear combination and Spatial Mismatch Index Model (SMI) to reveal the non-synergistic coupling relationship between multivariate and student-teacher rate in school districts, counties, and state multiple levels. The results suggest that compared to 10% of the spatial mismatch index at the state level, the proportion of mismatched areas at the school district and county levels was the same at 1%. NV, IN, VT, MA, and FL were mismatched at the state level but had good matches at the county and school district levels. Other unpredictable factors related to teacher shortages, such as workload, school rankings, and teacher vacancies, should be considered for further study in future research plans. This research provides valuable insights for policy interventions to improve the treatment of teachers in public school districts and promote educational equity.

**Keywords:** teacher shortage; geographic visualization of student-teacher rate; spatial mismatch index model

## 1. Introduction

The teacher shortage and its spatial and temporal dimensions have been a persistent problem in the education sector (Aragon 2016; Amanti 2019; Cowan et al. 2016). It is not just a problem of a shortage of educational resources in the labor market but also a problem of the spatial and temporal distribution of these resources, which affects student-teacher Rates and student achievement (Sutcher et al. 2019). Investigations at different scales, such as school, school district, county, state, and national levels, have shown spatial heterogeneity in the student-teacher rate, exposing various secular local education management deficiencies (Aragon 2016). Reichardt et al. (2020) mentioned that specific subject areas, grade level, and geographic location were the three main aspects of solving teacher shrinkage (Reichardt et al. 2020). In this research, we focused on visualizing the geographic heterogeneity of teacher shortages, which refers to mapping the trajectory of teacher shortages over time in counties and states. The student-teacher rate is used as a key indicator to measure the spatiotemporal disparity in teacher supply and demand (i.e., the number of students enrolled divided by the number of teachers from all sources who are willing and able to teach). By using geographic spatial visualization and a spatial-temporal mismatch model, this research aims to highlight the uneven distribution of educational resources and provide a new interdisciplinary paradigm known as educational geography. The spatial-temporal distribution of teacher supply and demand is useful to deepen the understanding

of educational inequality between students and teachers at the national level, to identify potential trends in teacher vacancies, and to induce labor markets to reallocate teachers to more efficient uses while mitigating teacher turnover.

Compared to teacher attrition, retention, and turnover in subject areas (i.e., STEM subjects) and teacher credentials or qualifications (Zweig et al. 2021), research on student-teacher Rates is not well documented. It dates back to the late 19th century (Lewit and Baker 1997) and focuses on class size reduction (Peers 2016; Jensen 2021; Solheim and Opheim 2019; Wang and Eccles 2016; Waasdorp et al. 2011; Finn et al. 2008) and teacher burnout (Jensen 2022; Borman and Dowling 2008; Jensen and Solheim 2020). Recent literature has highlighted high poverty in specific states such as Missouri (Reichardt et al. 2020), Mississippi Delta (Curran 2017), Arkansas (TNTP 2021), New York (Zweig et al. 2021), and professional teacher shortages such as music teachers (Hash 2021) and STEM teachers (Ridley-Kerr et al. 2020; Gross 2018; Woo 1985). Spatial-temporal descriptions of student-teacher rates are merely rare, not to mention macro-spectrum predictions of teacher shortages. Although Sutcher et al. (2019) proposed teacher mismatch between supply and demand as education reports, they did not mention their methods and navigate on large-scale assessment (Sutcher et al. 2019). Moreover, spatial mismatch originally refers to the phenomenon that the spatial allocation of production factors deviates from Pareto optimality due to various reasons, resulting in the loss of economic benefits (Kicsiny and Varga 2022; Li et al. 2022, Marrero-Vera et al. 2022). Previous spatial mismatch analysis intensively focuses on minor educator labor workforce allocation (Holzer 1991; Wasmer and Zenou 2002; Gobillon et al. 2007; Hsieh and Moretti 2019; Kicsiny and Varga 2022; Logan et al. 2020; Li et al. 2016; Silva et al. 2021; Wang et al. 2022; Yang et al. 2022). The spatial mismatch hypothesis is not employed in high-need school districts in the U.S. This paper took advantage of the theory of spatial mismatch and explored the components of student-teacher spatial mismatch so that teachers are optimally allocated.

Public school poverty, teacher comparable wages, the distance between school districts and highways, cities near school districts, and air quality were host factors related to student-teacher rate. Public school poverty is the free and reduced lunch program enrollment divided by the number of students enrolled in the same school, which directly reflects the household income of the students. Teacher-comparable wages are a critical factor in teachers' willingness to work in schools. The distance between school districts and highways is used to measure the degree of transportation convenience. The urban-rural factor is represented by the shortest distance of cities to which school districts are close, which is the core of the spatial mismatch hypothesis (i.e., teachers residing in inner cities face adverse labor market outcomes). (Zhang et al. 2007; Lau 2011; Li and Chu 2022). Air quality is an environmental factor that reveals the impact of the environment on the student-teacher rate. Five factors include socio-economic, environmental, and transportation comprehensive impacts on teacher shortage. There is insufficient evidence to support claims of a growing teacher shortage at the national level. If the teacher labor market is tight, it is more important than ever to ensure that students have access to quality education and achieve educational success. This study intends to either pinpoint the spatial mismatches areas, such as states and counties or suggest local education policymaking regarding increasing and decreasing teacher supply. By using a multi-criteria evaluation to calculate teacher demand, the mismatch index with the SMI model would automatically examine the coupling degree between the current supply student-teacher Rate and the integrated teacher demand degree.

## 2. Materials and Methods

### 2.1. Data

We captured the student-teacher rate of each school district and public-school poverty data from the Common Core Data (CCD). The representation of changes in teachers and students was done using panel data from 2015–2016 to the 2019–2020 school year, including cross-sectional and longitudinal two-sections. The CCD is the Department of

Education's primary database on public elementary and secondary education in the United States. It is a large-scale, cross-sectional, repeated survey with more than 55 states and 1,048,576 observations. The CCD consists of two components: the nonfiscal CCD and the fiscal CCD. The nonfiscal CCD is used for this research. Air quality data were from the U.S. Environmental Protection Agency (EPA). Comparable wages index for teachers were from the National Center for Education Statistics. U.S. highway and city data were from the U.S. Census Bureau. When developing the spatial mismatch model, we used data from the 2019–2020 school year as a case study to detect whether the spatial mismatch existed or not.

### 2.2. Study Framework

In order to estimate whether the current teacher labor supply satisfies potential teacher demand, we investigated public school poverty, air quality, teacher comparable wages, teacher transportation, and teachers residing close to cities' five aspects. Five steps were implemented in this research. First, we implemented an analytical hierarchy process (AHP) to identify the weights of five factors. Second, air quality at the county level was interpolated into school districts using the Kriging approach. We then conducted compromise programming (CP) to obtain the nearest school district surrounding U.S. primary and secondary roads and cities. Furthermore, we calculated a weighted linear combination based on the weights of five factors. Finally, we computed spatial mismatch indexes at county and state two levels according to each school districts' spatial mismatch indexes via the spatial mismatch model, as shown in Figure 1.

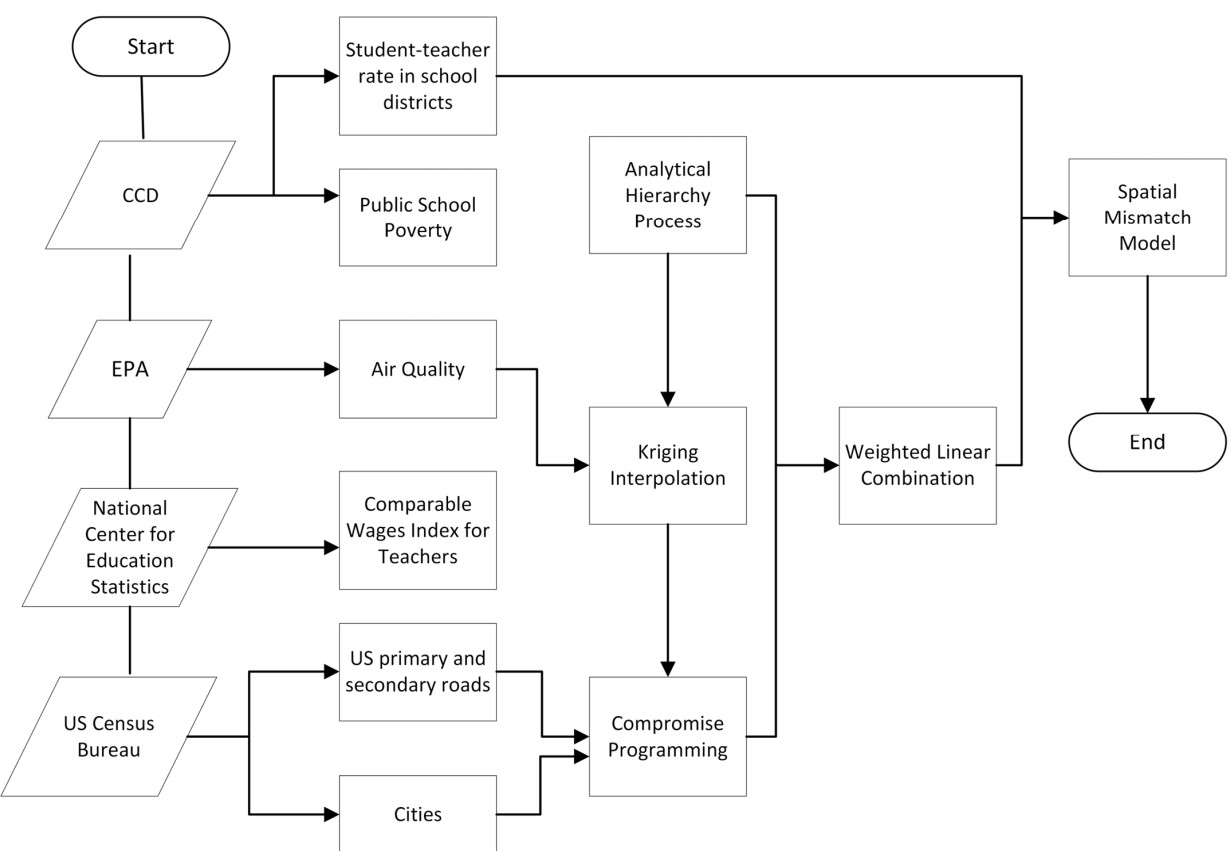

**Figure 1.** Study Framework.

### 2.3. Data Processing

We took two steps regarding data variability. For states without data, to keep the data real, we did not implement data imputation and count them as study objects. For the Analytical Hierarchy Process, we examined a survey of satisfaction with the educational environment and received a lot of random feedback. Using the feedback and the Delphi

method, the Education Policy Lab estimated the weights of each factor. There were some empty states in Figure 2 due to a lack of data. For states with data, we examined the normal distribution. If not qualified, we took the logarithm of a variable to ensure the consistency of the variable. Then, according to the empirical rule, we chose two standard deviations of the means as two cutoffs to eliminate outliers.

*2.4. Methods*

2.4.1. Analytical Hierarchy Process (AHP)

We used AHP to quantify expert input into weight assignments (Wind and Saaty 1980). First, factors were categorized into separate hierarchies based on similarity. Then, pair-wise comparison in Rates. After that, compute reciprocal, normalized, and priority vector matrices. Finally, verify internal consistency.

(1)  Use the Delphi method to determine the weight ($W$) of each factor by experts.
(2)  Identify the product ($M_i$) of each row in the matrix.

$$M_i = \prod_{j=1}^{n} a_{ij} \tag{1}$$

$a_{ij}$ is the proportion of the *i*th value divided by the *j*th value.

(3)  Calculate nth root of $M_i$.

$$W_i = \sqrt[n]{M_i} \tag{2}$$

(4)  Standardize the vector ($\vec{W_i}$).

$$\vec{W_i} = \frac{W_i}{\sum_{i=1}^{n} W_i} \tag{3}$$

(5)  Calculate the eigenvalue $\lambda_i$ of the matrix.

$$\lambda_i = \sum_{j=1}^{n} a_{ij} \vec{W_j} \tag{4}$$

(6)  Calculate the maximum of the eigenvalue $\lambda_i$

$$\lambda_{max} = \sum_{i=1}^{n} \frac{\lambda_i}{n \times W_i} \tag{5}$$

(7)  Calculate consistency value ($CI$).

$$CI = \frac{\lambda_{max} - n}{n - 1} \tag{6}$$

(8)  Implement the consistency test.

$$CR = \frac{CI}{RI} \tag{7}$$

If $CR$ is less than 0.1, it means that all weight ($W$) passes the consistency test; otherwise, it produces a new weight for each factor.

2.4.2. Compromise Programming (CP)

We used CP to minimize the distance to the ideal alternative (Zeleny 1982). First, we identified the best or worst value for each factor. Next, we determined weights and scaling coefficients ($p$). Eventually, we computed distance ($d$) with the following equation.

$$d_{x,y} = \left( \sum_{i}^{n} \frac{W_i}{\sum_{i}^{n} W_i} \left( \frac{Z_i^{best} - Z_{x,y,i}}{Z_i^{best} - Z_i^{worst}} \right)^p \right)^{\frac{1}{p}} \tag{8}$$

where $d_{x,y}$ denotes the distance. $W_i$ is the $i$th weights. $Z_i^{best}$ or $Z_i^{worst}$ means the $i$th factor maxim or minimum value. $Z_{x,y,i}$ represents the $i$th factor's value at the location of longitude $x$, latitude $y$.

### 2.4.3. Weighted Linear Combination

We used weighted linear combinations to maximize the suitability score based on a weighted sum of factor ratings. First, we define factor ratings ($F$). Then, we define weights ($W$). At last, we computed the suitability score ($S$) with the formula below.

$$S_{x,y} = \sum_i^n F_{x,y,i} \times W_i \tag{9}$$

where $S_{x,y}$ is the suitability score at the location of longitude $x$, latitude $y$. $F_{x,y,i}$ means the $i$th factor ratings. $W_i$ is the $i$th weights.

### 2.4.4. Spatial Mismatch Model

Based on the research findings of urban geography (Zhang et al. 2007), this paper constructed the spatial mismatch index model, which considers the discrepancy between the student-teacher ratio and the weighted linear combination of five factors: school poverty, air quality, comparable teacher wages, teacher transportation, and teacher proximity to urban areas, in order to explore the non-synergistic coupling law of their spatial distribution (Lau 2011; Li et al. 2013; Liu and Song 2005; Ma et al. 2012; Tang and Luo 2022). The spatial mismatch hypothesis posits that material constraints in high-need school districts impede the supply of teachers.

$$SMI = \frac{1}{2p} \times \left( \left| \frac{A_1}{A}p - p_1 \right| + \left| \frac{A_2}{A}p - p_2 \right| + \ldots + \left| \frac{A_i}{A}p - p_i \right| \right) = \frac{1}{2p} \sum_{i=1}^n \left| \frac{A_i}{A}p - p_i \right| \tag{10}$$

In the formula, the spatial mismatch index.
Where $p$ is the total amount of student-teacher rate ($t$ for the unit),
Additionally, $n$ is the number of school districts.
$p_i$ is the total student-teacher rate in a school district $i$ ($t$ for the unit).
$A$ refers to the comprehensive index (a certain region).
$A_i$ refers to the comprehensive index in the $i$th school district.
The higher the value, the more obvious the non-synergistic coupling will be, and vice versa.

## 3. Results

### 3.1. Student-Teacher Rate Description

3.1.1. Time Series Variability of Student-Teacher Rate

According to the longitudinal trend of the U.S. Rate in Table 1, the U.S. level score of student-teacher rates was 56.23 during 2016–2020. The average rate increased from 10.34 to 20.54 during 2016–2017. In 2018, the average rate bounced back to 14 after teacher supply. In 2019, the overall average student-teacher rate increased to 29.36. By 2020, the average rate dropped to 33.9. Furthermore, the trends of teacher supply decreased in recent years except in 2018 and 2020. During the five-year statistical process, high-need states did not constantly increase in a certain equal proportion. At the same time, the number of students increased in 2017 and 2020.

**Table 1.** Teacher, Student, and Student-teacher Rate at the state level.

| Name / Year | Student Numbers | | | | | Teacher Numbers | | | | | Student-Teacher Rate | | | | |
|---|---|---|---|---|---|---|---|---|---|---|---|---|---|---|---|
| | 2016 | 2017 | 2018 | 2019 | 2020 | 2016 | 2017 | 2018 | 2019 | 2020 | 2016 | 2017 | 2018 | 2019 | 2020 |
| Florida | 182,586.3 | 14,092,110.0 | 186,245.4 | 164,519.9 | 14,262,595.0 | 2,791,368 | 186,447.1 | 14,173,335 | 14,240,045 | 165,584.4 | 15.29 | 75.58 | 76.1 | 86.56 | 86.13 |
| Georgia | 111,653.2 | 8,821,730.0 | 115,799.6 | 116,932.5 | 8,844,565.0 | 1,727,085 | 114,531.9 | 8,843,210 | 8,836,010 | 117,567.3 | 15.47 | 77.02 | 76.37 | 75.57 | 75.23 |
| Hawaii | 11,746.9 | 907,750.0 | 12,033.5 | 12,132.1 | 905,440.0 | 181,995 | 11,781.7 | 904,185 | 906,390 | 12,220.8 | 15.49 | 77.05 | 75.14 | 74.71 | 74.09 |
| Idaho | 147.4 | 1,486,000.0 | 16,592.0 | 16,745.3 | 1,513,440.0 | 1179 | 16,203.9 | 1,505,930 | 1,550,220 | 16,790.3 | 8 | 91.71 | 90.76 | 92.58 | 90.14 |
| Indiana | 334.8 | 5,238,460.0 | 60,843.6 | 61,033.8 | 5,226,790.0 | 12,483 | 60,044.5 | 5,262,285 | 5,270,705 | 61,129.9 | 37.28 | 87.24 | 86.49 | 86.36 | 85.5 |
| Kansas | 1081.6 | 2,471,735.0 | 36,349.2 | 36,723.9 | 2,489,027.0 | 3594 | 36,193.3 | 2,484,294 | 2,488,665 | 36,449.1 | 3.32 | 68.29 | 68.35 | 67.77 | 68.29 |
| Kentucky | 42,671.7 | 3,420,085.0 | 42,064.2 | 41,826.9 | 3,459,650.0 | 686,252 | 42,028.7 | 3,404,890 | 3,389,105 | 42,223 | 16.08 | 81.37 | 80.95 | 81.03 | 81.94 |
| Louisiana | 18,476.7 | 3,580,620.0 | 40,234.9 | 38,909.2 | 3,551,105.0 | 214,238 | 48,405.2 | 3,574,780 | 3,555,270 | 38,585 | 11.6 | 73.97 | 88.85 | 91.37 | 92.03 |
| Maine | 702.7 | 901,925.0 | 14,637.2 | 14,908.0 | 897,930.0 | 7533 | 14,630.6 | 901,020 | 902,305 | 14,637.5 | 10.72 | 61.65 | 61.56 | 60.52 | 61.34 |
| Massachusetts | 803.0 | 4,822,664.0 | 73,381.6 | 73,868.8 | 0.0 | 15,207 | 72,413.6 | 4,770,373 | 4,758,597 | 0 | 18.94 | 66.6 | 65.01 | 64.42 | 0 |
| Michigan | 68,173.1 | 7,643,330.0 | 84,173.8 | 85,015.4 | 7,448,475.0 | 1,194,060 | 83,537.8 | 7,581,130 | 7,520,970 | 84,764.2 | 17.52 | 91.5 | 90.07 | 88.47 | 87.87 |
| Minnesota | 32,973.8 | 4,375,105.0 | 57,257.0 | 57,694.6 | 4,464,965.0 | 502,857 | 56,712.5 | 4,424,720 | 4,446,520 | 54,350.8 | 15.25 | 77.15 | 77.28 | 77.07 | 82.15 |
| Mississippi | 30,812.9 | 2,415,750.0 | 31,624.5 | 31,962.7 | 2,329,950.0 | 472,658 | 31,924.5 | 2,391,605 | 2,356,490 | 31,573.2 | 15.34 | 75.67 | 75.63 | 73.73 | 73.8 |
| Missouri | 4037.9 | 4,574,950.0 | 68,489.7 | 68,498.5 | 4,550,340.0 | 4675 | 67,926.2 | 4,577,060 | 4,567,205 | 68,678.3 | 1.16 | 67.35 | 66.83 | 66.68 | 66.26 |
| New Hampshire | 1284.0 | 895,625.0 | 14,637.4 | 14,631.5 | 883,955.0 | 42,176 | 14,806.8 | 897,779 | 890,565 | 14,689.1 | 32.85 | 60.49 | 61.33 | 60.87 | 60.18 |
| New Jersey | 246.3 | 7,034,195.0 | 115,342.1 | 116,185.1 | 6,991,850.0 | 5407 | 115,595.3 | 7,029,590 | 6,997,845 | 115,782.4 | 21.95 | 60.85 | 60.95 | 60.23 | 60.39 |
| New Mexico | 21,425.2 | 1,681,315.0 | 21,092.0 | 21,092.5 | 1,650,600.0 | 328,620 | 21,331 | 1,671,705 | 1,667,685 | 21,809.6 | 15.34 | 78.82 | 79.26 | 79.07 | 75.68 |
| New York | 49,154.9 | 13,648,880.0 | 213,158.9 | 212,088.6 | 13,229,940.0 | 611,619 | 209,151.3 | 13,623,315 | 13,498,660 | 208,947.3 | 12.44 | 65.26 | 63.91 | 63.65 | 63.32 |
| North Carolina | 80,572.4 | 7,750,310.0 | 100,400.8 | 100,220.3 | 7,405,010.0 | 1,260,022 | 100,219.6 | 7,767,565 | 7,762,485 | 95,898 | 15.64 | 77.33 | 77.37 | 77.45 | 77.22 |
| Ohio | 95,225.3 | 8,550,715.0 | 98,658.9 | 101,739.4 | 6,699,300.0 | 1,633,156 | 102,484.2 | 8,521,995 | 8,478,810 | 79,910.6 | 17.15 | 83.43 | 86.38 | 83.34 | 83.83 |
| Oklahoma | 6347.1 | 3,469,515.0 | 41,528.5 | 42,384.0 | 0.0 | 21,092 | 41,022.4 | 3,475,460 | 3,494,455 | 0 | 3.32 | 84.58 | 83.69 | 82.45 | 80.86 |
| Pennsylvania | 112,008.1 | 8,645,580.0 | 122,065.7 | 123,348.4 | 8,139,920.0 | 1,639,451 | 122,677.9 | 8,644,745 | 8,653,785 | 113,575 | 14.64 | 70.47 | 70.82 | 70.16 | 71.67 |

**Table 1.** *Cont.*

| Name / Year | Student Numbers | | | | | Teacher Numbers | | | | | Student-Teacher Rate | | | | |
|---|---|---|---|---|---|---|---|---|---|---|---|---|---|---|---|
| | **2016** | **2017** | **2018** | **2019** | **2020** | **2016** | **2017** | **2018** | **2019** | **2020** | **2016** | **2017** | **2018** | **2019** | **2020** |
| Rhode Island | 10,404.4 | 710,750.0 | 10,653.0 | 10,710.1 | 717,630.0 | 138,475 | 10,639.7 | 714,745 | 717,180 | 10,653.6 | 13.31 | 66.8 | 67.09 | 66.96 | 67.36 |
| South Dakota | 7169.4 | 680,675.0 | 9831.6 | 9865.4 | 698,425.0 | 99,282 | 9772 | 687,645 | 693,355 | 9915.8 | 13.85 | 69.66 | 69.94 | 70.28 | 70.44 |
| Tennessee | 63,408.0 | 5,007,810.0 | 64,019.4 | 64,116.0 | 4,434,947.0 | 992,324 | 64,270.3 | 5,009,835 | 5,031,545 | 56,659.1 | 15.65 | 77.92 | 78.25 | 78.48 | 78.27 |
| Texas | 337,650.2 | 26,804,245.0 | 358,100.9 | 100,021.8 | 9,952,305.0 | 5,240,665 | 353,561.4 | 27,006,705 | 7,791,975 | 130,397.7 | 15.52 | 75.81 | 75.42 | 77.9 | 76.32 |
| Utah | 171.3 | 3,299,005.0 | 0.0 | 0.0 | 0.0 | 1403 | 0 | 3,341,376 | 0 | 0 | 0 | 0 | 0 | 0 | 0 |
| Vermont | 7842.4 | 19,029.0 | 3213.5 | 0.0 | 0.0 | 73,596 | 415.1 | 192,116 | 0 | 0 | 0 | 0 | 0 | 0 | 0 |
| Washington | 800.4 | 0.0 | 0.0 | 0.0 | 0.0 | 50,191 | 0 | 0 | 0 | 0 | 0 | 0 | 0 | 0 | 0 |
| Wisconsin | 45,555.3 | 0.0 | 0.0 | 0.0 | 0.0 | 690,363 | 0 | 0 | 0 | 0 | 0 | 0 | 0 | 0 | 0 |
| U.S. Virgin Islands | 545.0 | 0.0 | 0.0 | 0.0 | 0.0 | 6559 | 0 | 0 | 0 | 0 | 0 | 0 | 0 | 0 | 0 |
| Arkansas | 0.0 | 845.0 | 46.0 | 4.0 | 900.0 | 0 | 3 | 895 | 905 | 4 | 0 | 281.67 | 19.46 | 226.25 | 225 |
| Delaware | 0.0 | 681,320.0 | 9398.7 | 9623.6 | 0.0 | 0 | 9208.2 | 681,465 | 692,025 | 0 | 0 | 73.99 | 72.51 | 71.91 | 0 |
| District of Columbia | 0.0 | 427,213.0 | 6599.8 | 7300.8 | 0.0 | 0 | 6667.5 | 436,122 | 458,044 | 0 | 0 | 64.07 | 66.08 | 62.74 | 0 |
| Illinois | 0.0 | 10,108,540.0 | 127,935.1 | 132,175.8 | 9,706,040.0 | 0 | 127,261.3 | 10,024,440 | 9,836,054 | 132,463.5 | 0 | 79.43 | 78.36 | 74.42 | 73.27 |
| Iowa | 0.0 | 2,549,155.0 | 35,292.3 | 35,357.1 | 2,586,620.0 | 0 | 35,538.6 | 2,559,250 | 2,574,165 | 35,473.3 | 0 | 71.73 | 72.52 | 72.8 | 72.92 |
| Maryland | 0.0 | 4,435,680.0 | 60,234.3 | 60,710.5 | 4,547,020.0 | 0 | 59,762.8 | 4,472,380 | 4,484,300 | 61,484.6 | 0 | 74.22 | 74.25 | 73.86 | 73.95 |
| Montana | 0.0 | 731,875.0 | 10,497.6 | 10,576.2 | 660,190.0 | 0 | 10,536.2 | 733,287 | 738,545 | 9195.9 | 0 | 69.46 | 69.85 | 69.83 | 71.79 |
| Nebraska | 0.0 | 1,595,970.0 | 23,703.0 | 23,911.6 | 1,632,080.0 | 0 | 23,542.8 | 1,618,830 | 1,631,960 | 23,535 | 0 | 67.79 | 68.3 | 68.25 | 69.35 |
| Nevada | 0.0 | 2,368,723.0 | 23,709.0 | 23,240.0 | 2,495,328.0 | 0 | 23,704.7 | 2,448,875 | 2,481,122 | 25,466.5 | 0 | 99.93 | 103.29 | 106.76 | 97.98 |
| North Dakota | 0.0 | 548,515.0 | 8988.9 | 9469.1 | 579,290.0 | 0 | 8956.1 | 559,600 | 569,225 | 9242 | 0 | 61.24 | 62.25 | 60.11 | 62.68 |
| Oregon | 0.0 | 2,894,565.0 | 29,822.8 | 30,055.2 | 2,902,234.0 | 0 | 29,664.3 | 2,903,243 | 2,914,929 | 29,770.2 | 0 | 97.58 | 97.35 | 96.99 | 97.49 |
| South Carolina | 0.0 | 3,857,038.0 | 52,466.8 | 52,729.5 | 3,930,864.0 | 0 | 50,789.4 | 3,887,745 | 3,904,036 | 53,450.5 | 0 | 75.94 | 74.1 | 74.04 | 73.54 |

### 3.1.2. Spatial Distribution of Student-Teacher Rate

County-Level Distribution of Student-Teacher Rate

County-level distributions were scattered across the years. In the 2015–2016 school year, we found six school districts from five counties in different states with a student-teacher rate of more than 100. In terms of the number of school districts, Merrimack County in New Hampshire had the highest teacher shortage with a rate of 250.56, as shown in Table 2. This was followed by Rogers County in Oklahoma with a Rate of 235.83, Thurston County in Nebraska with a Rate of 165.66, Thurston County in Washington with a Rate of 165.66, and Elkhart County in Indiana with a Rate of 157.32. For the 2016–2017 school year, 21 school districts in seven counties in five states are highlighted for having a Rate greater than 100. Union County in Arkansas had the highest rate of 281.67. Carson City in Nevada had the second-highest rate at 127.35. Interestingly, Coos County in Oregon and Manistee County had the same number of seven school districts over the rate of 105. In the 2017–2018 school year, eight school districts from four counties in four states are noted. White Pine County in Nevada had the highest rate of 148.85, but Comal County in Texas had the lowest rate of 106.8, which is beyond 100. For the 2018–2019 school year, thirteen school districts from six counties in six states are noted. Hidalgo County in Texas had the highest rate of 257.43. The Rate of Elko County in Nevada and Union County in Arkansas was over 200. For the 2019–2020 school year, 29 school districts from six counties in five states are highlighted. The highest rate of 283.11 was located in Olmsted County in Minnesota, which covered 10 school districts.

**Table 2.** The top five counties with the highest student-teacher rates during the research time.

| State | Year | County | School District Count | Student | Teacher | Rate (Stu/Tea) |
|---|---|---|---|---|---|---|
| Oklahoma | 2015–2016 | Rogers | 1 | 1356 | 5.75 | 235.83 |
| Nebraska | 2015–2016 | Thurston | 1 | 2195 | 13.25 | 165.66 |
| Indiana | 2015–2016 | Elkhart | 1 | 5322 | 33.83 | 157.32 |
| New Hampshire | 2015–2016 | Merrimack | 2 | 5813 | 23.2 | 250.56 |
| Washington | 2015–2016 | Thurston | 1 | 2195 | 13.25 | 165.66 |
| Nevada | 2016–2017 | Carson City | 2 | 193,735 | 1521.32 | 127.35 |
| Oregon | 2016–2017 | Coos | 7 | 50,660 | 472.67 | 107.18 |
| Indiana | 2016–2017 | Owen | 1 | 13,650 | 126.29 | 108.08 |
| Oregon | 2016–2017 | Josephine | 2 | 54,050 | 508.58 | 106.28 |
| Indiana | 2016–2017 | Owen | 1 | 13,650 | 126.29 | 108.08 |
| Michigan | 2016–2017 | Manistee | 7 | 27,740 | 248.06 | 111.83 |
| Arkansas | 2016–2017 | Union | 1 | 845 | 3 | 281.67 |
| Texas | 2017–2018 | Comal | 2 | 160,275 | 1500.7 | 106.8 |
| Oregon | 2017–2018 | Wheeler | 3 | 5080 | 45.14 | 112.54 |
| Nevada | 2017–2018 | White Pine | 1 | 9775 | 65.67 | 148.85 |
| Michigan | 2017–2018 | Arenac | 2 | 10,140 | 93.3 | 108.68 |
| Texas | 2018–2019 | Hidalgo | 5 | 244,380 | 949.31 | 257.43 |
| New Mexico | 2018–2019 | San Miguel | 2 | 15,255 | 92.3 | 165.28 |
| Nevada | 2018–2019 | White Pine | 1 | 8275 | 49 | 168.88 |
| Louisiana | 2018–2019 | Plaquemines Parish | 2 | 24,805 | 189 | 131.24 |

**Table 2.** *Cont.*

| State | Year | County | School District Count | Student | Teacher | Rate (Stu/Tea) |
|---|---|---|---|---|---|---|
| Nevada | 2018–2019 | Elko | 2 | 50,965 | 209 | 243.85 |
| Arkansas | 2018–2019 | Union | 1 | 905 | 4 | 226.25 |
| Minnesota | 2019–2020 | Olmsted | 10 | 126,790 | 447.84 | 283.11 |
| Oregon | 2019–2020 | Wheeler | 3 | 8090 | 64.33 | 125.76 |
| Minnesota | 2019–2020 | Goodhue | 6 | 34,035 | 459.09 | 74.14 |
| Indiana | 2019–2020 | Johnson | 6 | 138,660 | 1152.76 | 120.29 |
| South Dakota | 2019–2020 | Davison | 3 | 16,445 | 229.18 | 71.76 |
| Arkansas | 2019–2020 | Union | 1 | 900 | 4 | 225 |

Note: Teacher numbers that have decimals indicate substitute teachers were converted into formal teachers.

State-Level Distribution of Student-Teacher Rate

Considering Figure 2 of the spatial investigation, it can be seen that in 2015–2016, Indiana (Rate: 37.28), New Hampshire (Rate: 32.84), Massachusetts (Rate: 18.94), and Washington (Rate: 62.71) were the top four shortage states in terms of student-teacher rate. Meanwhile, in 2016–2017, the top three states include Idaho (Rate: 91.71), Michigan (Rate: 91.5), and Arkansas (Rate: 281.67). In 2017–2018, four states had the highest student-teacher Rates, including Indiana (Rate: 90.76), Michigan (Rate: 90.07), Nevada (Rate: 103.29), and Oregon (Rate: 97.35). In 2018–2019, the top four states with teacher shortages were Idaho (Rate: 92.58), Louisiana (Ratio: 91.37), Arkansas (Rate: 226.25), and Nevada (Rate: 106.76). In 2019–2020, there were five states with high-need teachers, including Oregon (Rate: 97.49), Nevada (Rate: 97.98), Arkansas (Rate: 225), Louisiana (Rate: 92.03), and Idaho (Rate: 90.14). In 2020–2021, the three states with the greatest need for teachers are Louisiana (Rate: 20.35), Oregon (Rate: 19.44), and Nevada (Rate: 18.76).

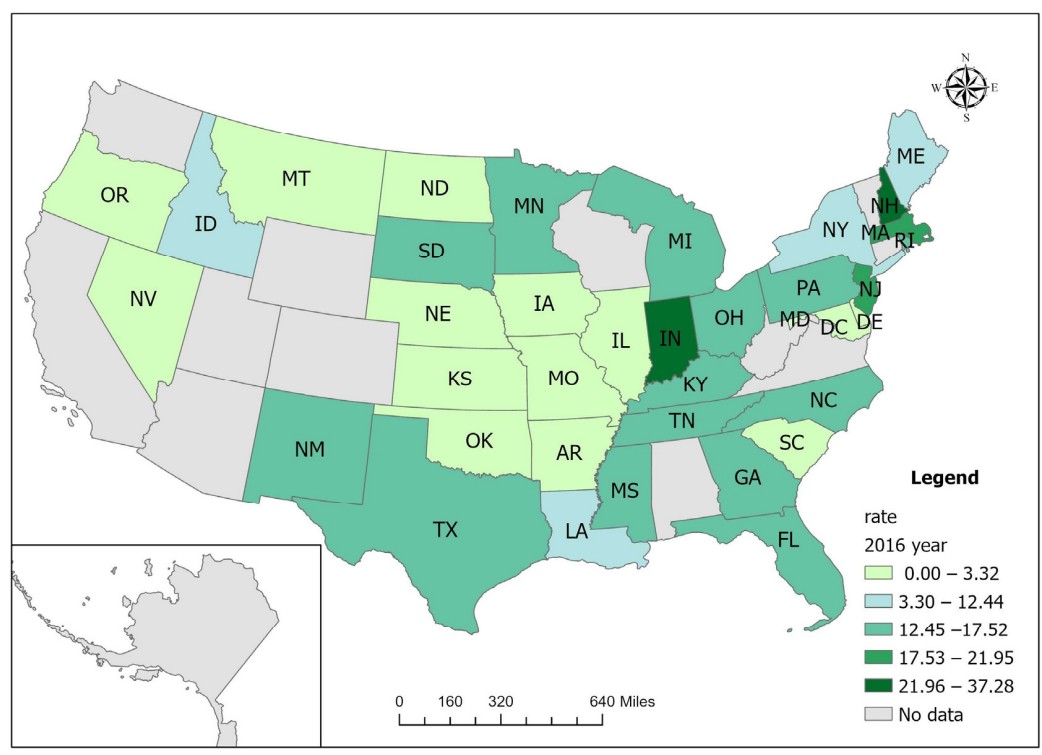

**Figure 2.** *Cont.*

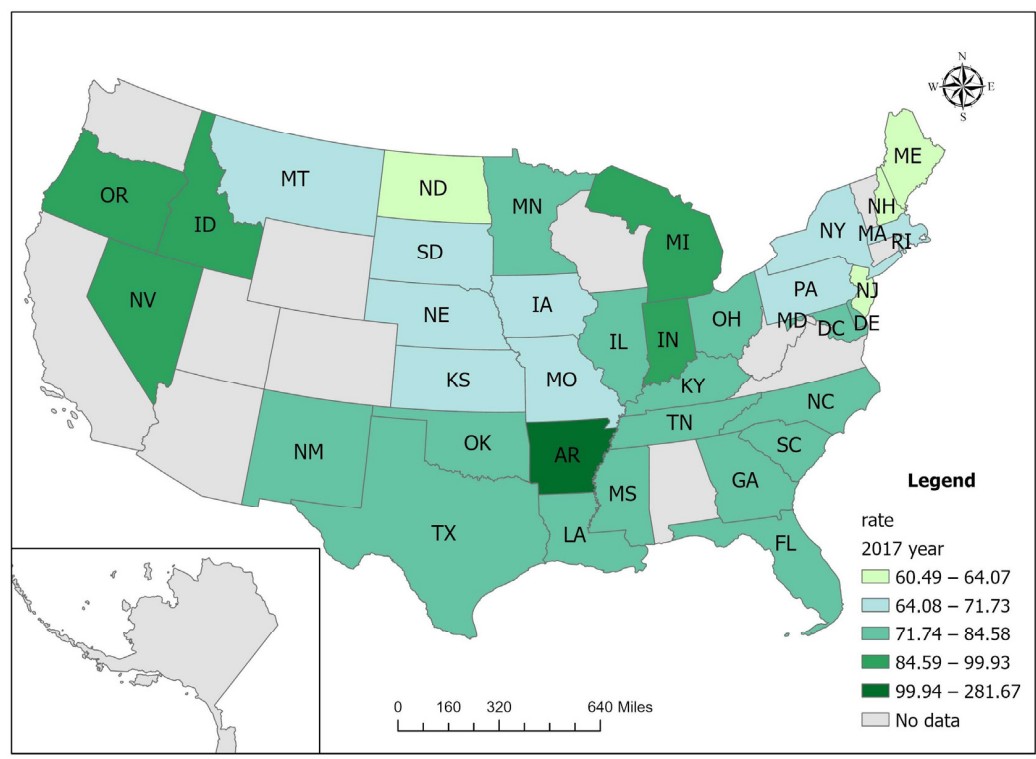

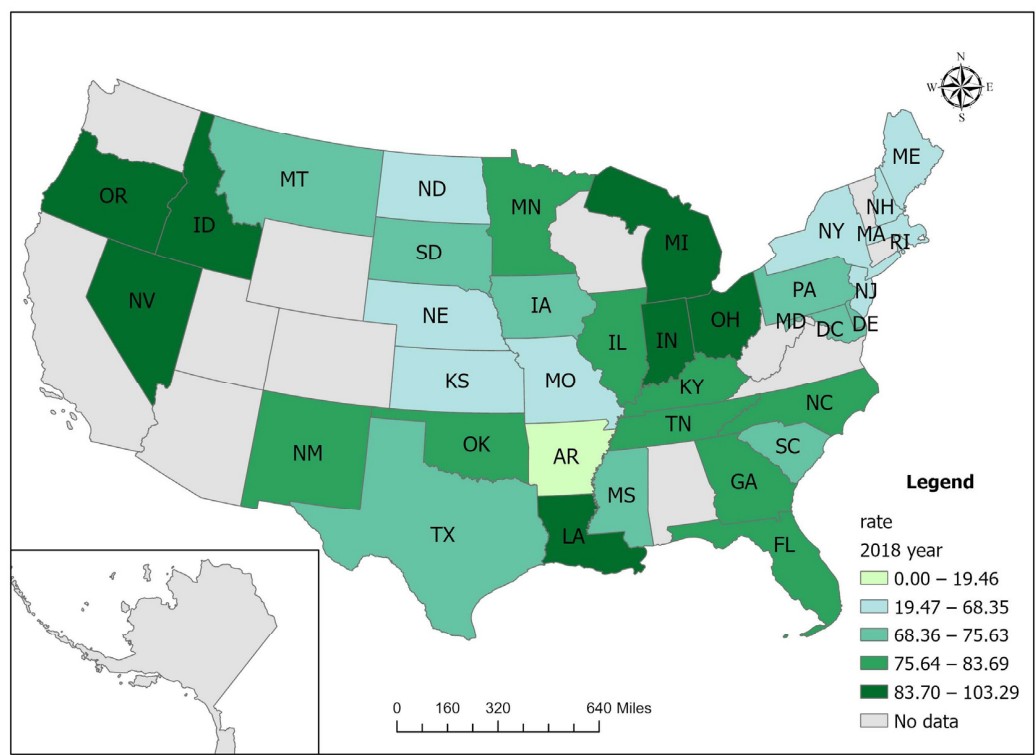

**Figure 2.** *Cont.*

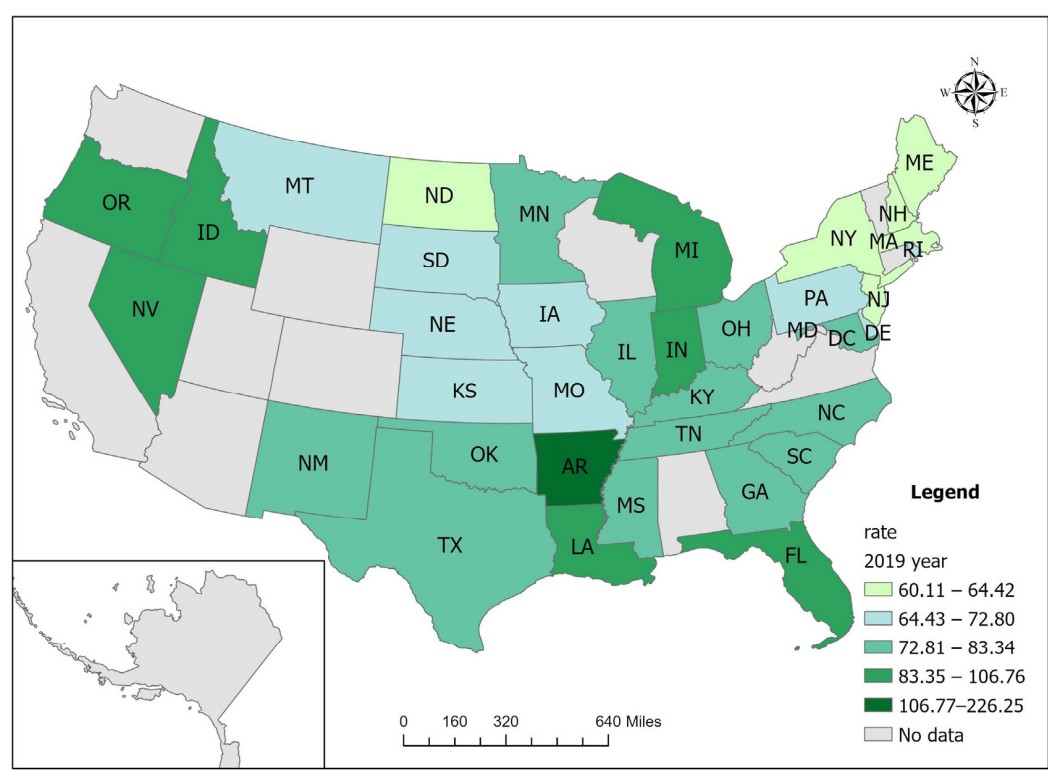

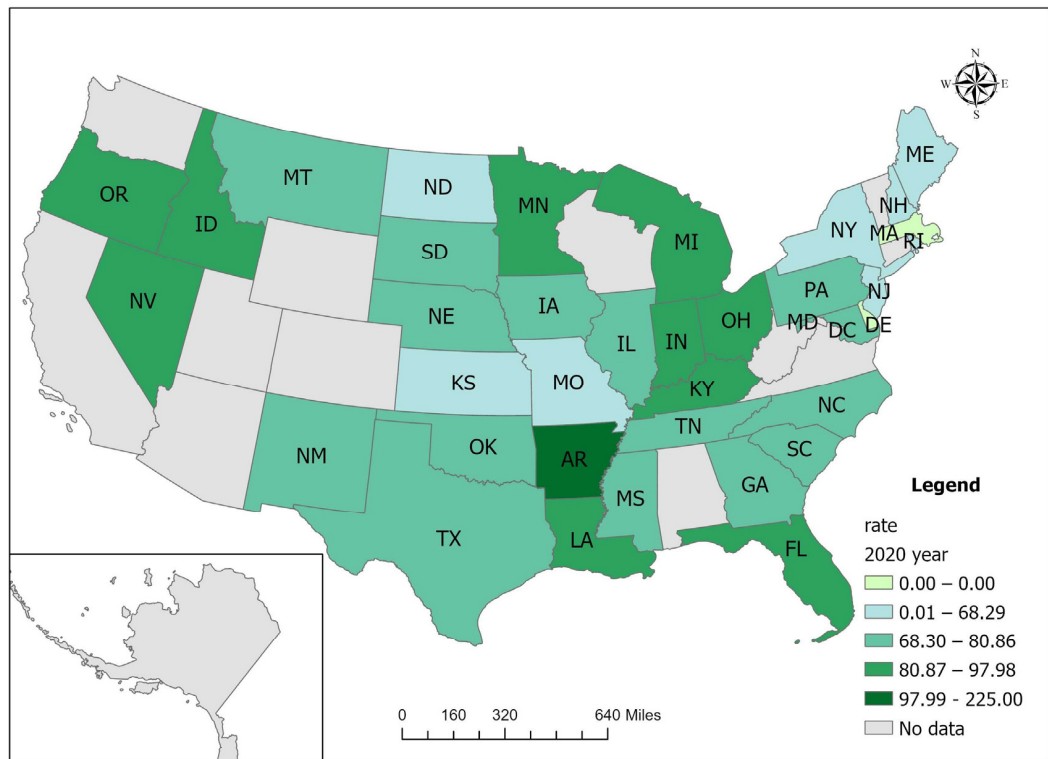

**Figure 2.** 2016–2020 Student-Teacher Rate Distribution in the U.S.

*3.2. AHP Results*

We generated the reciprocal matrix (W) in Table 3, the normalized matrix in Table 4, and the weighted vector in Table 5.

**Table 3.** Reciprocal Matrix Table.

| Factors | Air Quality | Highway | City | School Poverty | Salary |
|---|---|---|---|---|---|
| Air Quality | 1.00 | 0.50 | 0.33 | 0.25 | 0.20 |
| Highway | 2.00 | 1.00 | 0.50 | 0.33 | 0.25 |
| City | 3.00 | 2.00 | 1.00 | 0.50 | 0.33 |
| School Poverty | 4.00 | 3.00 | 2.00 | 1.00 | 0.50 |
| Comparable wage | 5.00 | 4.00 | 3.00 | 2.00 | 1 |
| Sum | 15.00 | 10.50 | 6.83 | 4.08 | 2.28 |

Note: The air quality weights were related to other factors based on the Delphi method by experts. The remaining weights were derived from the air quality weights.

**Table 4.** Normalized Matrix Table.

| Factors | Air Quality | Highway | City | School Poverty | Salary |
|---|---|---|---|---|---|
| Air Quality | 0.067 | 0.048 | 0.049 | 0.061 | 0.088 |
| Highway | 0.133 | 0.095 | 0.073 | 0.082 | 0.109 |
| City | 0.200 | 0.190 | 0.146 | 0.122 | 0.146 |
| School Poverty | 0.267 | 0.286 | 0.293 | 0.245 | 0.219 |
| Comparable wage | 0.333 | 0.381 | 0.439 | 0.490 | 0.438 |
| Sum | 1.00 | 1.00 | 1.00 | 1.00 | 1.00 |

Note: The air quality weight corresponding to the air quality of 0.067 is equal to the air quality weight of 1 (Table 3) divided by the sum of 15 (Table 3). The rest was obtained in the same way.

**Table 5.** Weighted Vector Table.

| Factors | Air Quality | Highway | City | School Poverty | Salary |
|---|---|---|---|---|---|
| Air Quality | 0.062 | 0.049 | 0.054 | 0.065 | 0.083 |
| Highway | 0.125 | 0.099 | 0.081 | 0.087 | 0.104 |
| City | 0.187 | 0.197 | 0.161 | 0.131 | 0.139 |
| School Poverty | 0.250 | 0.296 | 0.322 | 0.262 | 0.208 |
| Comparable wage | 0.312 | 0.394 | 0.483 | 0.524 | 0.416 |

Note: The air quality of 0.062 was equal to the air quality weight in Table 3, times the average of the first row in Table 4. The rest was obtained in the same way.

According to Table 6, the CR value (i.e., 0.0258) was less than 0.1. It means the weight of each factor is plausible.

**Table 6.** Consistency Test Table.

| Factors | Priority Vector (1) | Weighted Sum Vector (2) | Consistency Vector (1)/(2) | λ max | CI | RI | CR |
|---|---|---|---|---|---|---|---|
| Air Quality | 0.062 | 0.314 | 5.035 | | | | |
| Highway | 0.099 | 0.495 | 5.023 | | | | |
| City | 0.161 | 0.815 | 5.060 | | | | |
| School Poverty | 0.262 | 1.337 | 5.108 | | | | |
| Comparable wage | 0.416 | 2.129 | 5.115 | | | | |
| λ max | | | | 5.115 | | | |
| CI | | | | | 0.029 | | |
| RI | | | | | | 1.12 | |
| CR | | | | | | | 0.0258 |

Note: The air quality's priority vector of 0.062 was equal to the average of the first row in Table 4. The air quality weighted sum vector was equal to the sum of the first row in Table 5. λ max is the maximum of the consistency vector. CI derived from the Formula (6). CR is derived from the Formula (7).

*3.3. CP Results*

In light of the CP calculation principle, highway, and city variables were implemented "Euclidean distance", "zonal statistic as table", and "raster calculator" functions; the minimum, maximum, and weights of all variables were shown in Table 7.

**Table 7.** Parameters of compromise programming in locating school districts.

| Criteria | Minimum | Maximum | AHP Weight |
|---|---|---|---|
| Air Quality | 65.389 | 153.705 | 0.062 |
| Highway | 0 | 211.26 | 0.098 |
| City | 0 | 7.00 | 0.161 |
| School Poverty | 0 | 1.00 | 0.261 |
| Comparable wage | 0.65 | 1.35 | 0.416 |

(1) Air Quality Results

In light of air quality on the county level in the U.S. in 2019, we took advantage of the Kriging method to interpolate air quality indexes into school districts with raster calculations. The highest air quality was located in the northwest U.S., while the lowest air quality was located in the west U.S., as shown in Figure 3. The higher the air quality, the better the places where teachers were willing to live.

(2) Highway Results

In the highway raster results of Figure 4, we found that Alaska had the maximum value due to highway scarcity, whereas the minimum values were in the southeast U.S., which has a high density of highways. The more convenient transportation is, the better place teachers prefer to be.

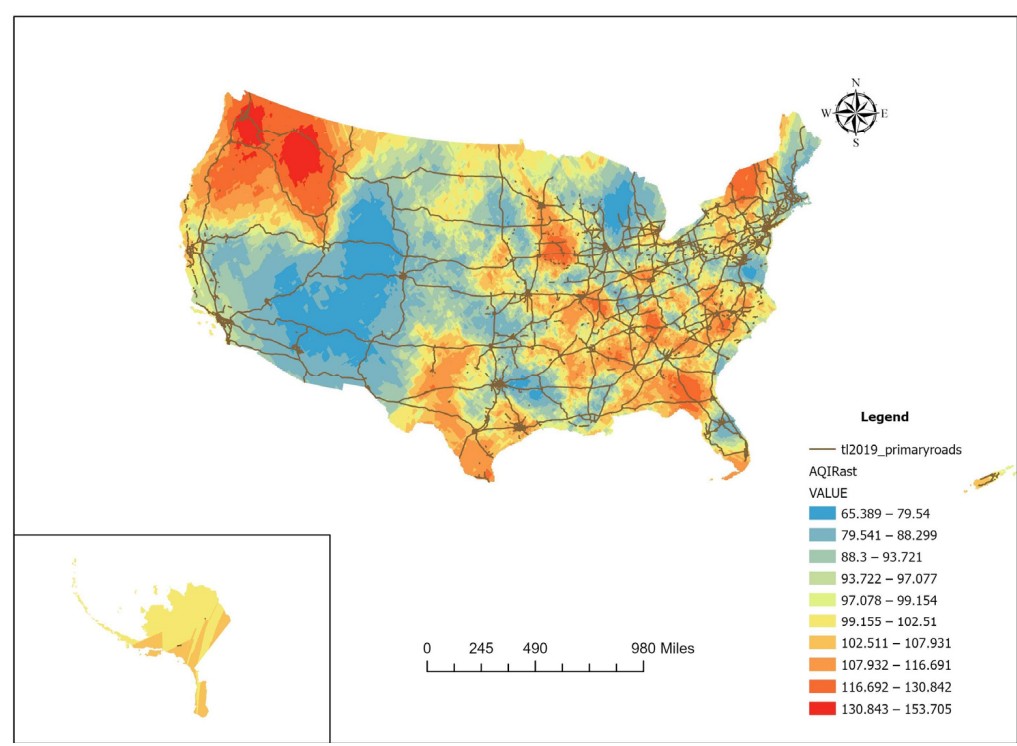

**Figure 3.** Air Quality Interpolation Results.

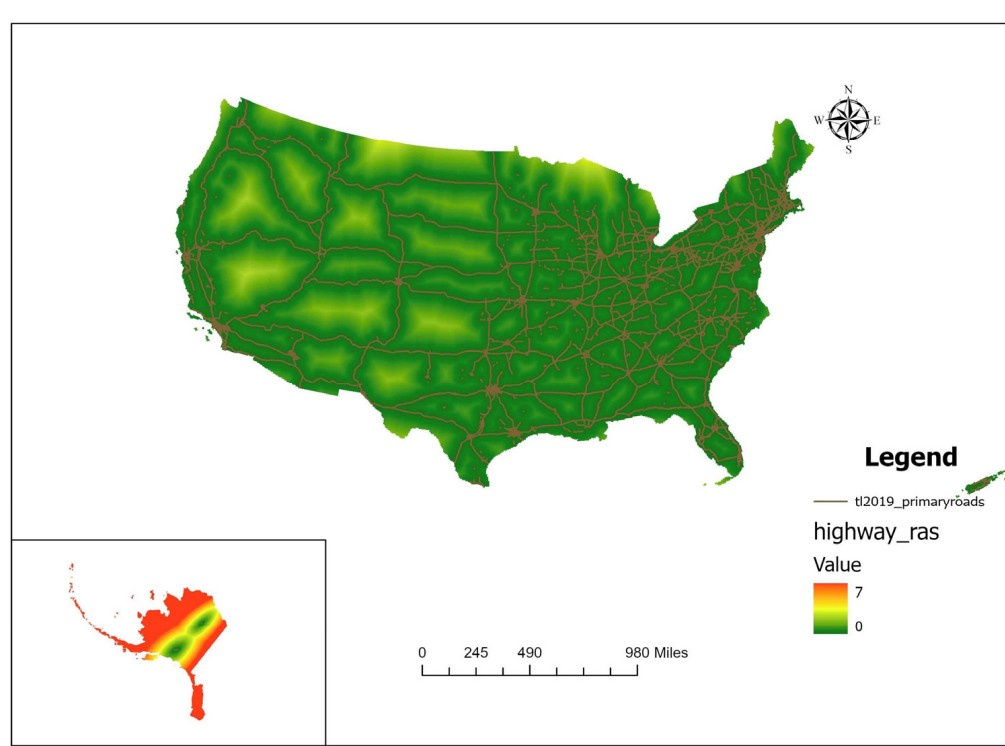

**Figure 4.** Compromise Programming Results of Highways.

(3)    Public School Poverty Results

In Figure 5, U.S. public school poverty was classified into five levels, including very low poverty, low poverty, medium poverty, high poverty, and very high poverty. Very low poverty was less than 0.17, low poverty was between 0.17 and 0.38, medium poverty was between 0.38 and 0.57, high poverty was between 0.57 and 0.79, and very high poverty was above 0.79. The highest public-school poverty was in Alaska and the coastal areas of the

U.S., i.e., they were underdeveloped economic regions. The lowest public-school poverty was in the eastern U.S., which is at a developed economic level.

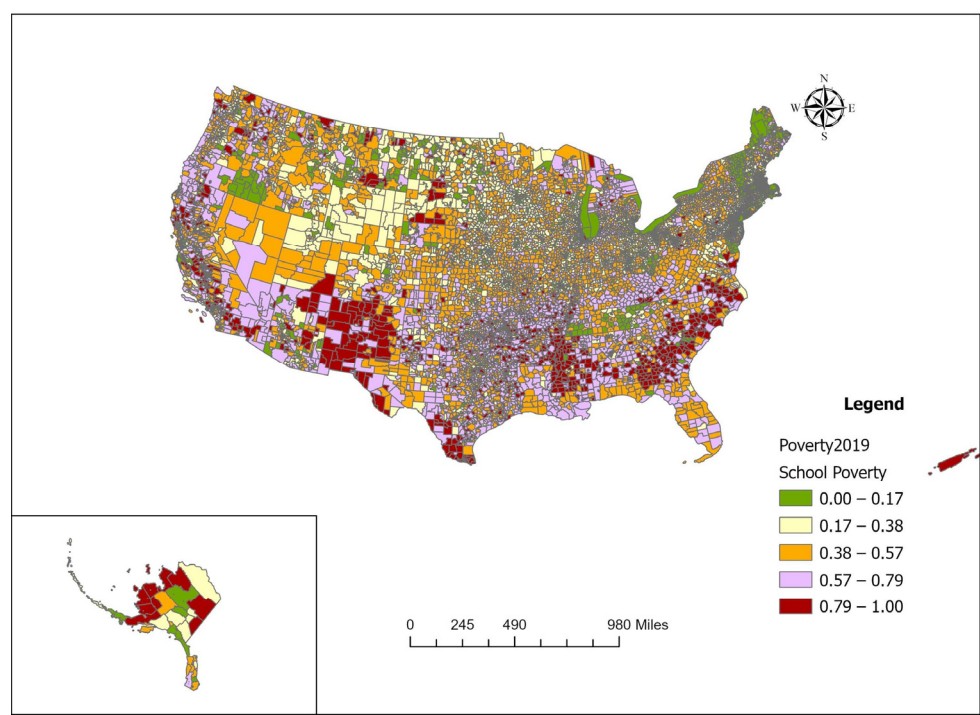

**Figure 5.** Public School Poverty Normalization in U.S. Public School Districts.

(4)    City Results

In Figure 6 of the U.S. city distribution, the closest school district to the city had the minimum value, while the furthest school districts from the city had the maximum value in the raster map. Alaska had the most distant school district from the city.

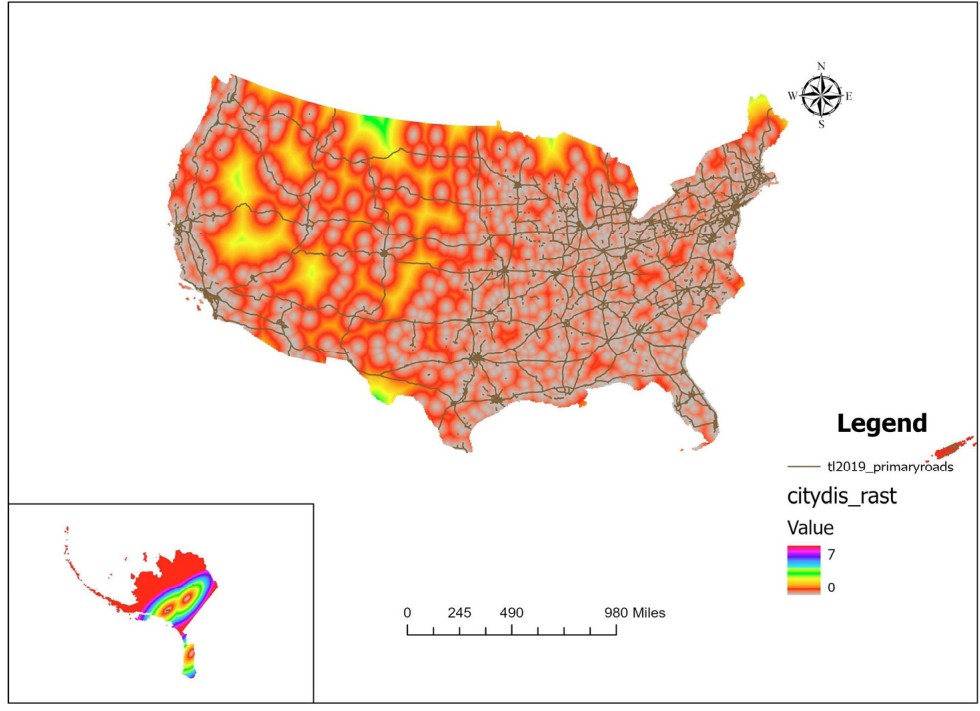

**Figure 6.** Compromise Programming Results of Cities.

(5)    Teacher Comparable Wages Results

Figure 7 is the teacher comparable wage normalization in U.S. 12,833 public school districts. The high teacher comparable wage index was located in 2691 school districts in the Middle East of the U.S. At the same time, 2519 school districts were ranked as having the low teacher comparable wage index in the U.S.

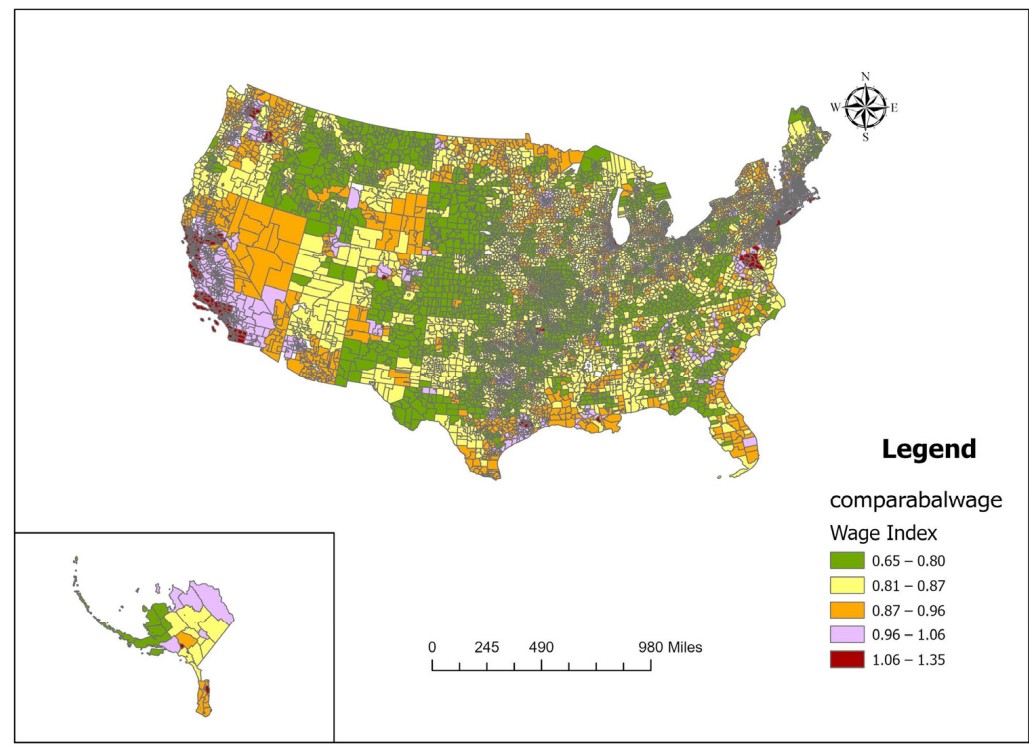

**Figure 7.** Teacher Comparable Wages Normalization in U.S. Public School Districts.

*3.4. Weighted Linear Combination Results*

Figure 8 is presented according to the distribution of the comprehensive index at the U.S. public school district level. Public school districts were divided into five categories. From the lowest comprehensive index to the highest comprehensive index, 12,833 school districts were classified in the range of 0.29–0.58, 0.581–0.67, 0.671–0.73, 0.731–0.79, and 0.791–0.96. There were 1557 school districts at the range of 0.791–0.96, 3944 school districts at the range of 0.731–0.79, 3743 school districts at the range of 0.671–0.73, 2286 school districts at the range of 0.581–0.67, and 1303 school districts at the range of 0.29–0.58.

*3.5. SMI of Individual School Districts*

Figure 9 is presented according to the distribution of the Spatial Mismatch Index (SMI) at the U.S. public school district level. Public school districts were divided into five categories. From the lowest SMI to the highest SMI, 12,833 school districts were classified as good match, good match, match, low match, and no match. There were 229 school districts at no match level, 1002 school districts at low match level, 3047 school districts at match level, 6593 school districts at good match level, and 1962 school districts at very good match level. Non-match school districts took up 1.78% of total school districts.

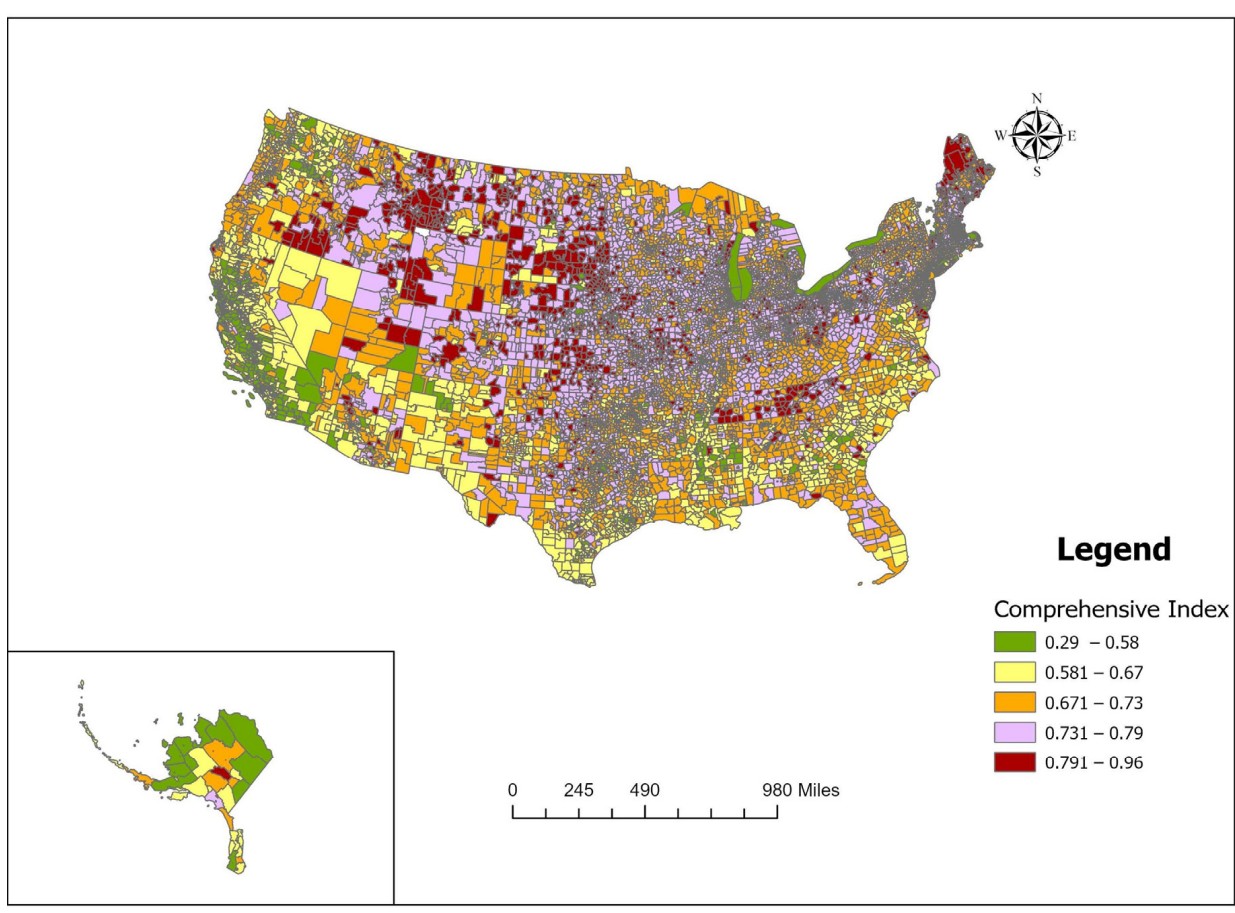

**Figure 8.** Comprehensive Index Distribution at U.S. Public School Districts Level.

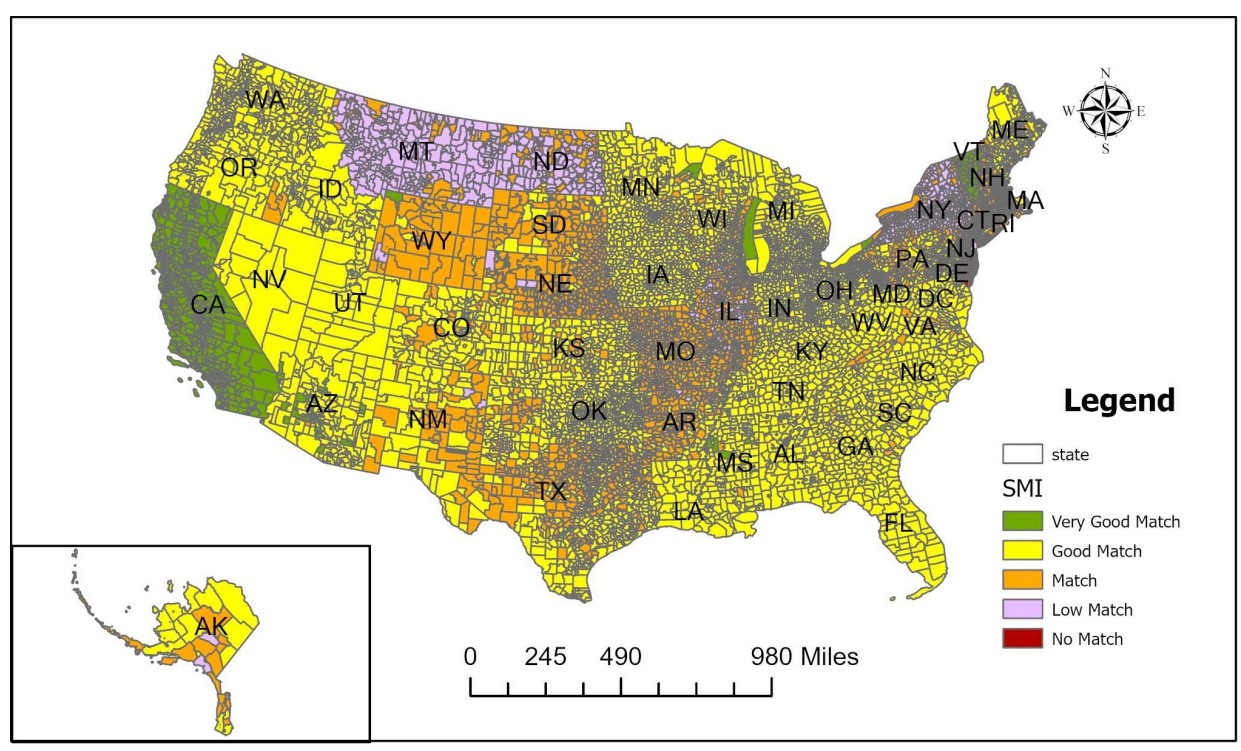

**Figure 9.** The Spatial Mismatch Index (SMI) at the U.S. public school district.

*3.6. Results of the SMI Models*

3.6.1. The SMI on the State Level

According to the spatial mismatch index distribution at the U.S. state level in Figure 10, there are explicitly significant results in the spatial mismatch model at the state level (i.e., SMI). The preliminary characteristics include: (1) Most SMI was less than 0.5, meaning they have a good match between teacher supply and teacher demand. (2) California has the very best match among states, and Arizona is the second-best match state. (3) Only a few states were no-match, such as MV, IN, VT, MA, and FL, owing to SMI beyond 0.5. (4) Non-match states took up 10% of total states.

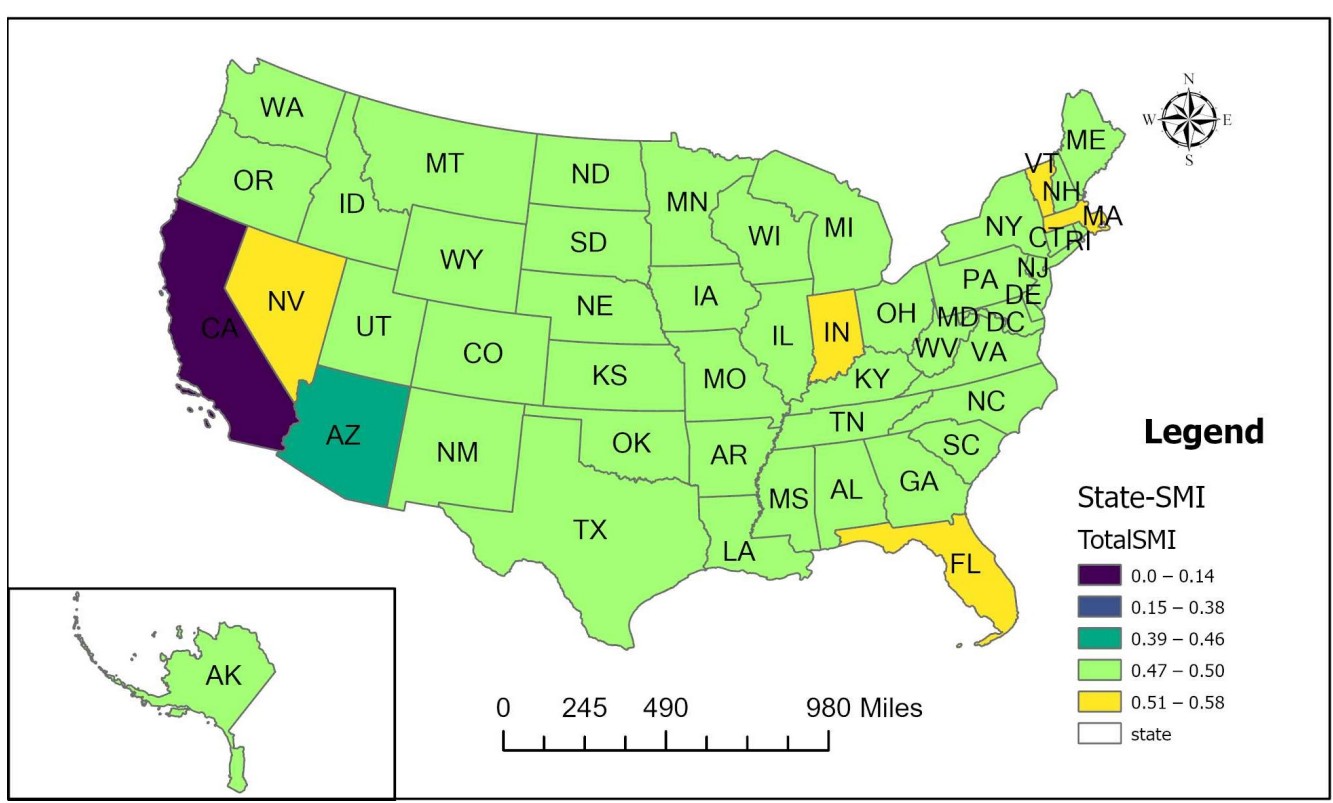

**Figure 10.** Spatial Mismatch Index Distribution at the U.S. State Level.

3.6.2. The SMI Mismatch at the County Level

According to the spatial mismatch index distribution map at the county level in Figure 11, the following results are generated. SMI at a county level was classified into five levels, including very good-match, good-match, match, low-match, and non-match. In total 3233 counties, there were 2610 counties at very good match level (i.e., SMI is no more than 0.1), 382 counties at good match level (i.e., SMI is less than 0.2), 130 counties at match level (i.e., SMI is less than 0.35.), 75 counties at low match level (i.e., SMI was less than 0.5.), and 36 counties (i.e., SMI is more than 0.51) at non-match level. Non-match counties occupied 1.1% of total counties.

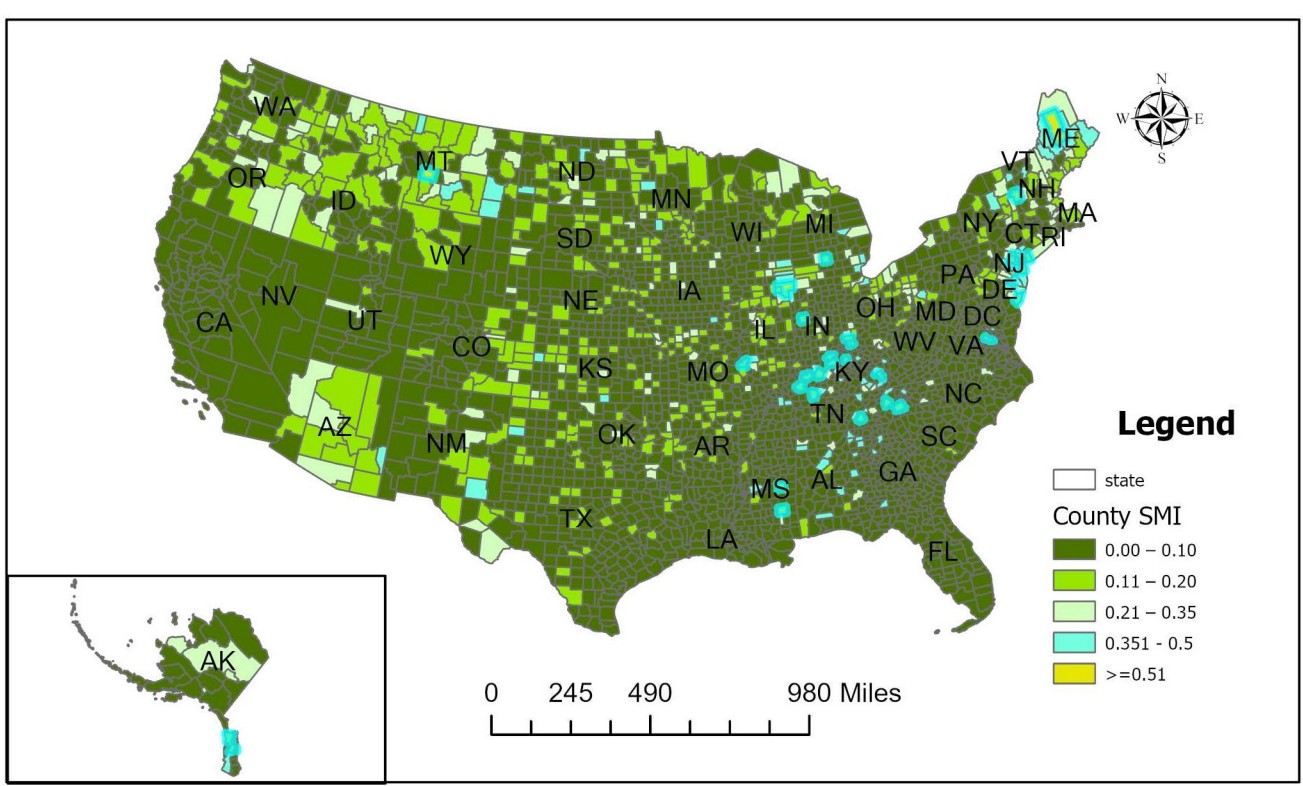

**Figure 11.** Spatial Mismatch Index Distribution at U.S. County Level.

## 4. Discussion

Based on the aforementioned mismatch between teacher supply and demand at multiple scales, we found several compelling implications. First, although the proportion of mismatched areas at the school district and county levels was the same at 1%, meaning that teacher supply satisfied potential teacher demand at two levels, there was apparently spatial heterogeneity: NV, IN, VT, MA, and FL were mismatched at the state level but had good match at the county and school district levels. In other words, the differentiation was masked at the inter-county level. Second, the spatial mismatch index model in this research, which examines the uncoupled relationship between student-teacher ratios and a weighted linear combination of five factors: school poverty, air quality, comparable teacher salaries, teacher transportation, and teacher proximity to urban areas, provided a new sight for state-level analysis. Thirdly, in light of five selected factors, we failed to find an imbalance between teacher supply and teacher demand. There may be other unpredictable factors that trigger teacher shortages, such as workload, school rankings, and teacher vacancies. These should be considered for further investigation in the next research plans.

In fact, the government is sparing no effort to improve the treatment of teachers and alleviate the potential shortage of teachers via proper policy direction and perfecting incentives. For example, some significant funding has been provided by nonprofit educational institutions, such as the American Association for the Advancement of Science (AAAS), which is dedicated to promoting and recruiting STEM majors and professionals to become K–12 teachers. The Robert Noyce Teacher Scholarship Program is reportedly designed to increase the number of K–12 teachers with strong STEM backgrounds who are willing to teach in high-need school districts (http://nsfoyce.org, accessed on 31 January 2020). Throughout 2021, the Noyce Project has supported 1083 Noyce Scholarship programs with a total award amount of $1,251,686,486.00. In Montana, for example, Salish Kootenai College, Montana State University, the University of Montana, and the University of Providence received $9,506,316.00 in awards for six programs from 2009 to 2019, aligning faculty resources and alleviating some of the pressure on the faculty labor market. At the same time, the existing hierarchical education system pays more attention to advanced education

(college education or above) and ignores general education (senior high school or below). This may cause a temporary dynamic imbalance between teachers and students. In general, general education has a greater impact on culture and society as students navigate life beyond their undergraduate experience (Smith and Tarantino 2019).

Moreover, school reputation is seen as a key barrier that explains how public schools are affected by school choice and competition (Jenkins 2020). All teachers really believe that their benefits are closely related to the school's reputation. There are two polarizations between enrollment above demand and enrollment below demand. Some prestigious urban schools have excellent teachers and students with intense competition, but rural schools have extreme shortages of teachers based on population. In addition, school administration has an impact on teacher turnover. A teaching environment is a blend of social, emotional, and instructional elements that stimulate teachers' subjective willingness and career aspirations. Creating a welcoming teacher education program is essential to teacher retention (Menzies 2023).

Beyond the spatial mismatch analysis, we are also aware that the deep root of the mismatch is insufficient financial security and political competition. Financial security is not enough to estimate the workload and salary increase in teachers, which caused the mass exodus. First, there are intangible extra burdens that interfere with teachers' normal routines, such as reading academies, grading, and extra reading training. Teachers are expected to sacrifice their leisure time to perform these extra tasks. Second, teachers' benefits lack an inflationary budget. In almost all school districts, teachers' salaries are only budgeted at a minimum of 3% of income. When the inflation rate exceeds 3%, the cost of living rises, which eats up the teacher salary increase. If there is no proper teacher financial stimulation regulation to guarantee teacher development, it is hard to solve the teacher shortage naturally. As for policy issues, some states have addressed the teacher shortage in a positive way, but some states have hidden it. For example, in Texas, in order to slow down the deterioration of the teacher shortage. On 11 March 2022, the government established a task force to address the teacher shortage. On 25 July 2022, new laws will be constructed to help Texas schools unintentionally contribute to the teacher shortage. In contrast, ten other states are missing data in the CCD. One of the main reasons may be that state education agencies are unwilling to share their teacher and student information to avoid negative repercussions.

Some limitations should be noted as a result of modeling the spatial mismatch between teachers and students. First, this research does not account for data imputation. Inevitably, there is missing data in several states. In order to ensure the authenticity of the data, we did not use data imputation methods to fill in the missing data, so the missing data does not affect our results. Second, the statistical unit in this research is a school district, not an individual school. Since school-level computation is too large for traditional computers to perform smoothly, we chose the school district as the research unit after empirical school-level data merging failure in Python. To refine our data further, delving into individual schools should examine our results.

## 5. Conclusions

The research identified the impact of teacher salary, school poverty, transportation, and environmental factors on the student-teacher ratio but did not consider student-learning outcomes. It does not mean that student learning outcomes are insignificant; it is a key criterion for assessing educational equity. However, the student learning outcome is the final result that several factors related to teachers contributed and depends on individual learning preference, practice time, and intelligence development. Herein, student learning outcomes were not selected in this research. Furthermore, it also measured the presence of geographic disparities in the U.S. in terms of teacher-student mismatch across school districts, counties, and states. We developed a spatial mismatch model to quantify these disparities with spatial mismatch indices. It could be valuable in addressing and reducing educational disparities by providing a more transparent and data-driven understanding

of how educational resources are distributed geographically. In addition, the research suggests that these disparities are shaped by factors such as state and local regulation and social service provision rather than federal policy. By identifying areas of greatest disparity, policymakers and educators may be able to target resources and interventions where they are most needed. It is, therefore, a good reference for local policymakers.

**Author Contributions:** X.W.: conceptualization, writing—original draft, formal analysis, resources, data curation, investigation, and visualization. J.Z.: conceptualization, methodology, software, formal analysis, validation, writing—review and editing, and supervision. All authors have read and agreed to the published version of the manuscript.

**Funding:** This research received no external funding.

**Institutional Review Board Statement:** Not applicable.

**Informed Consent Statement:** Not applicable.

**Data Availability Statement:** The datasets used during the current study are available from the corresponding author upon reasonable request.

**Acknowledgments:** The literature review and data source for this research is partially supported by Feng Li's NSF. Thanks for her unconditional help. We are also grateful for the contributions of two anonymous reviewers.

**Conflicts of Interest:** The authors declare no conflict of interest.

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
