# Peer review of "Case Study on Spatial Mismatch between Multivariate and Student-Teacher Rate in U.S. Public School Districts"

_socsci, doi:10.3390/socsci13020093_

Round 1

Reviewer 1 Report

Comments and Suggestions for Authors

Just one suggestion. 

The measurement of individual poverty is one of the most complex tasks for statistical research. In this article, reference is made to Public-school poverty, based on "The nonfiscal CCD is used for this research".

For a non-US reader it would be useful to have some methodological information on this data.

It would then be interesting in the conclusions to explain more clearly whether your research has achieved relevant results also with respect to the geographical distribution of the quality of education received by students. To conclude, a question: why were student learning outcomes not taken into consideration? In any case, the research is of great interest

Author Response

Dear Editors,

    Thank you so much for giving us the opportunity to revise the manuscript (Submission ID socsci-2753690). Please extend our thanks to the three anonymous reviewers for their valuable suggestions and comments. We have reviewed these comments carefully and have made revisions accordingly. Based on the reviewers’ suggestions and comments. For your reading convenience, we colored the suggestions and comments from the reviewers in blue.

Sincerely yours,

The measurement of individual poverty is one of the most complex tasks for statistical research. In this article, reference is made to Public-school poverty, based on "The nonfiscal CCD is used for this research".

It is an honor to accept your comments. Thanks for your understanding.

For a non-US reader it would be useful to have some methodological information on this data.

A good thinking. Thanks for your consideration.

It would then be interesting in the conclusions to explain more clearly whether your research has achieved relevant results also with respect to the geographical distribution of the quality of education received by students. To conclude, a question: why were student learning outcomes not taken into consideration? In any case, the research is of great interest.

We really appreciate the mention. We have already added some interpretations to the conclusion section on Line 419-425, “The research identified the impact of teacher salary, school poverty, transportation, and environmental factors on the student-teacher ratio, but did not consider student learning outcomes. It does not mean that student learning outcome is insignificant, it is a key criterion for assessing educational equity. However, student learning outcome is the final results that several factors related to teachers contributed and depends on individual learning preference, practice time and intelligence development. Herein, student learning outcomes were not selected in this research.”

Reviewer 2 Report

Comments and Suggestions for Authors

Dear Authors,

The main question addressed by the manuscript refers to the spatial dynamics of student- STEM teacher balance in public schools in the US.

This topic is important at meso (at the county and school district) levels, and makes an addition at macro (state) level. 

Conclusions can be completed with recommendations for local policy makers.

Bibliographic references should be supplemented with the educational importance of the topic (what is the connection between the topic and educational equity), and should be added some longitudinal statistical data.   

I suggest, changing Figures 2,3,4 in the table and introducing the US level scores. 

Tables 2,3,4,5 need a more detailed explanation.

Bibliographies need some editing. 

Author Response

Dear Editors,

    Thank you so much for giving us the opportunity to revise the manuscript (Submission ID socsci-2753690). Please extend our thanks to the three anonymous reviewers for their valuable suggestions and comments. We have reviewed these comments carefully and have made revisions accordingly. Based on the reviewers’ suggestions and comments. For your reading convenience, we colored the suggestions and comments from the reviewers in blue.

Sincerely yours,

Dear Authors,

The main question addressed by the manuscript refers to the spatial dynamics of student- STEM teacher balance in public schools in the US.

An excellent conclusion. Thanks for your comments.
This topic is important at meso (at the county and school district) levels, and makes an addition at macro (state) level. 

That is a substantial interpretation. A special thank from us to you.

Conclusions can be completed with recommendations for local policy makers.

Thanks for your important suggestions. We added the sentence for local policy makers to the section of conclusion on Line 433-434.

Bibliographic references should be supplemented with the educational importance of the topic (what is the connection between the topic and educational equity), and should be added some longitudinal statistical data.   
 That is an important point. Thanks for your attention. I have changed References into the supplement section and added longitudinal statistical data there.

I suggest, changing Figures 2,3,4 in the table and introducing the US level scores. 

Thanks for your good suggestions. We accepted them and reorganized them into Table 1 on Line 200. And we added the sentence “the U.S. level score of student-teacher rates was 56.23 during 2016-2020.” on Line 190-191.

Tables 2,3,4,5 need a more detailed explanation.

That is a good idea. Thanks for the mention. We added notes at the bottom of Tables 2, 3, 4, and 5. Please see Line 242-243, 247-248, 250-251, and 256-258.

Bibliographies need some editing. 

Thanks for your suggestions. We edited the Bibliographies part on Line 436-536.

Reviewer 3 Report

Comments and Suggestions for Authors

The author(s) depicted the interplay between student-teacher equilibrium and various factors (i.e., teacher salary, school poverty, transportation, and environmental conditions) over a period of 5 years (2015-2020). In general, the manuscript is well-structured and written, and the use of Analytic Hierarchy Process (AHP), Compromise Programming (CP), weight linear combination, and Spatial Mismatch Index (SMI) Model in this investigation is both logical and appropriate. I believe that the approach and results highlighted in this article could serve as a valuable source of insight for education administrators, policy makers, and researchers. I have a few minor recommendations for improving the manuscript, which are outlined below.

Page 3, line 122: It would benefit readers to have detailed information regarding the process and individuals involved in the expert inputs for the AHP. Additionally, please review the citation for Saaty (1980). It appears to be referencing "Wind, 1980" as listed in the reference list.

Page 4, line 166 – Page 5, line 169: The sentence ‘Based on the research findings of urban geography (Zhang et al., 2007), this paper constructed the spatial mismatch index model between student-teacher Rate and school poverty to explore the non-synergistic coupling law of their spatial distribution (LAU, 2011; Li et al., 2013)’ is confusing. According to the study framework, Spatial Mismatch Index considers the discrepancy between the student-teacher ratio and the weighted linear combination of five factors: school poverty, air quality, comparable teacher wages, teacher transportation, and teacher proximity to urban areas. This index takes into account more than just school poverty in evaluating the balance between students and teachers.

Page 15, lines 342 - 345: The authors make a valid point about the inherent bias in aggregate data and the potential for statistical errors due to the ecological fallacy. I concur that this limitation should be addressed in the discussion. However, given its apparent nature, is it necessary to extensively discuss the first limitation as described on Page 15, lines 337-342? In the meantime, I am wondering if the Author(s) have any suggestions for more suitable methods for state-level analysis. If possible, it would greatly benefit the readers if some information about these methods could be provided.

Author Response

Dear Editors,

    Thank you so much for giving us the opportunity to revise the manuscript (Submission ID socsci-2753690). Please extend our thanks to the three anonymous reviewers for their valuable suggestions and comments. We have reviewed these comments carefully and have made revisions accordingly. Based on the reviewers’ suggestions and comments. For your reading convenience, we colored the suggestions and comments from the reviewers in blue.

Sincerely yours,

The author(s) depicted the interplay between student-teacher equilibrium and various factors (i.e., teacher salary, school poverty, transportation, and environmental conditions) over a period of 5 years (2015-2020). In general, the manuscript is well-structured and written, and the use of Analytic Hierarchy Process (AHP), Compromise Programming (CP), weight linear combination, and Spatial Mismatch Index (SMI) Model in this investigation is both logical and appropriate. I believe that the approach and results highlighted in this article could serve as a valuable source of insight for education administrators, policy makers, and researchers. I have a few minor recommendations for improving the manuscript, which are outlined below.

We appreciate your thoughtful summary and thank you for all minor suggestions.

Page 3, line 122: It would benefit readers to have detailed information regarding the process and individuals involved in the expert inputs for the AHP. Additionally, please review the citation for Saaty (1980). It appears to be referencing "Wind, 1980" as listed in the reference list.

Thanks for your comments. We added AHP details on Line 122-125. Thanks for correcting the citation for Saaty (1980). We added Wind name on the citation (see Line 132).

Page 4, line 166 – Page 5, line 169: The sentence ‘Based on the research findings of urban geography (Zhang et al., 2007), this paper constructed the spatial mismatch index model between student-teacher Rate and school poverty to explore the non-synergistic coupling law of their spatial distribution (LAU, 2011; Li et al., 2013)’ is confusing. According to the study framework, Spatial Mismatch Index considers the discrepancy between the student-teacher ratio and the weighted linear combination of five factors: school poverty, air quality, comparable teacher wages, teacher transportation, and teacher proximity to urban areas. This index takes into account more than just school poverty in evaluating the balance between students and teachers.

That is an excellent point. Thank you for pointing that out. We revised the sentence into “This paper constructed the spatial mismatch index model, which considers the discrepancy between the student-teacher ratio and the weighted linear combination of five factors: school poverty, air quality, comparable teacher wages, teacher transportation, and teacher proximity to urban areas, to explore the non-synergistic coupling law of their spatial distribution.” on Line 170-175.

Page 15, lines 342 - 345: The authors make a valid point about the inherent bias in aggregate data and the potential for statistical errors due to the ecological fallacy. I concur that this limitation should be addressed in the discussion. However, given its apparent nature, is it necessary to extensively discuss the first limitation as described on Page 15, lines 337-342? In the meantime, I am wondering if the Author(s) have any suggestions for more suitable methods for state-level analysis. If possible, it would greatly benefit the readers if some information about these methods could be provided.

This is another important point. We appreciate your discovery. We revised “the inherent bias in aggregate data and the potential for statistical errors due to the ecological fallacy “ into “the spatial mismatch index model in this research, which examines the uncoupled relationship between student-teacher ratios and a weighted linear combination of five factors: school poverty, air quality, comparable teacher salaries, teacher transportation, and teacher proximity to urban areas, provided a new sight for state-level analysis.” on Line 359-363.